# The Behavior Mechanism of the Urban Joint Distribution Alliance under Government Supervision from the Perspective of Sustainable Development

**Na Zhang** [1,2,*] **, Xiangxiang Zhang** [1] **and Yingjie Yang** [2]

1    School of Economics and Management, Shihezi University, Shihezi 832000, China;
     zhangxiangxiangjan@163.com
2    Institute of Artificial Intelligence, De Montfort University, Leicester LE1 9BH, UK; yyang@dmu.ac.uk
*    Correspondence: zhangnanuaa@163.com

**Abstract:** Urban joint distribution is closely related to the national economy and people's livelihood, and governments and enterprises play an active role in the process of urban joint distribution. From the perspective of government regulations, this paper explores the mechanism and evolution law of the behavior of an urban joint distribution alliance. Based on the evolutionary game theory, a model of homogeneous enterprises participating in urban joint distribution operations under the guidance of government regulations is constructed. The mechanism and follow-up of alliance behavior are analyzed through the simulation of the relationship between parameters. It is found that, firstly, from the perspective of government regulations, in the early stage of the implementation of urban joint distribution projects, when the benefits of synergetic cooperation of enterprise alliances are relatively low and the costs are relatively high, it is necessary for the government to formulate incentive policies to improve government subsidies or to increase the penalties for non-cooperation of enterprises; Once a benign logistics environment and market mechanism are formed, the cooperation benefits increase, and the costs decrease, the government can then withdraw its supervision. Secondly, in the process of establishing urban joint distribution alliance under government supervision, it is better for the enterprises to actively achieve alliance cooperation and obtain government subsidies instead of passively accepting government supervision and paying penalties, in order to promote the formation of logistics ecological environment and market mechanism.

**Keywords:** government regulation; urban joint distribution; distribution alliance; behavior generation mechanism

## 1. Introduction

As a new distribution mode, urban joint distribution has the characteristics of low cost, high efficiency and environmental friendliness, which has become an inevitable trend of urban logistics distribution development [1–7]. Therefore, the construction of urban modern logistics system with joint distribution as the core is of great significance to promote the sustainable development of urban modern logistics. In 2018, the total amount of social logistics in China was 283.1 trillion RMB, and the total revenue was 10.1 trillion RMB, while the total cost of social logistics was up to 13.3 trillion RMB, indicating that China's logistics scale is large, but the cost is also relatively high, resulting in great pressure on the final zero kilometers of terminal distribution. Since 2011, the state has issued a number of normative documents related to urban joint distribution to solve terminal distribution problems (as shown in Table 1), and carried out urban joint distribution pilot projects in 22 cities including Nanjing, Wuhan, Xiamen and Chengdu to actively explore and innovate joint distribution models.

**Table 1.** National-level regulations and documents on urban joint distribution since 2011.

| Year | Normative Documents | Contents Related to Urban Joint Distribution |
|---|---|---|
| 2011 | Opinions of the General Office of the State Council on Policies and Measures to Promote the Healthy Development of the Logistics Industry | Study and formulate urban distribution management measures according to the principle of law, high efficiency and environmental protection. |
| 2012 | Opinions of the State Council on Deepening the Reform of Circulation System and Accelerating the Development of Circulation Industry | Support circulation enterprises in building modern logistics centers and actively develop unified distribution. |
| 2013 | Notice of the General Office of the Ministry of Finance and the Ministry of Commerce on Organizing the Application for the Pilot Project of Urban Joint Distribution | Innovate the mode of joint distribution and collaborative distribution, and build a reasonably distributed, orderly and green urban distribution service system. |
| 2014 | Medium and Long Term Development Plan of Logistics Industry 2014–2020 | Encourage alliance cooperation in transportation, postal, trade, supply and marketing, and publication sales, integrate and utilize existing logistics resources, and improve joint distribution capacity. |
| 2018 | Three-year Action Plan on Transport Restructuring 2018–2020 | Guide mega-city clusters and regional central cities in planning and building green freight distribution networks, and improve the construction of urban distribution network nodes and supporting facilities for docking and loading and unloading distribution vehicles. |
| 2019 | Notice of the General Office of the Ministry of Commerce on Replicating and Promoting the Pilot Experience of Urban Joint Distribution | Since the pilot project of urban joint distribution was launched, local governments have actively explored and innovated modes of joint distribution, and built a service system of urban joint distribution with reasonable layout, orderly operation and environmental protection. |

Urban joint distribution has also gradually attracted the attention of the academic community, which spurs multi-field and multi-perspective research. The research mainly falls into two categories. On the one hand, some research works focus on the optimization of urban joint distribution routes: Yan et al. [8] constructed an optimization model of multi-center collaborative distribution vehicle routing to solve the problem of urban collaborative distribution vehicle routing under current e-commerce conditions. Groß et al. [9] explored an efficient, economical and reliable route optimization method to optimize the route problem of urban logistics vehicles by constructing a cargo delivery terminal to satellite location planning model. Sun et al. [10,11] constructed a mathematical model for integrated vehicle routing problem based on joint distribution strategy through planning and computer algorithm to deal with the logistics distribution problem of rural e-commerce. Mario et al. [12] proposed a novel dynamic programming method to classify the major problems of urban joint distribution and obtain the optimal route of urban freight distribution in the management of urban logistics by using an accurate algorithm. In particular, reinforcement learning based on multi-agent simulation-adaptive dynamic programming to evaluate joint delivery systems in uncertain environments and large neighborhood search algorithms can effectively deal with vehicle routing problems related to travel time and travel time rules [13,14]. On the other hand, there are analyses highlighting the effectiveness

and mode of urban joint distribution: Song [15], Duan [16] empirically analyzed the effectiveness of rural e-commerce joint distribution mode from the perspective of Shared logistics. Sullet [17] pointed out that the urban joint distribution helped to reduce road traffic congestion, which can significantly improve the urban environment. Therefore, local governments need to develop the unity of the urban and regional policy and freight system, and promote the development of city logistics [18], especially implementing warehousing and distribution patterns for medium and small e-commerce enterprises to promote the integration of retail e-commerce logistics distribution resources [19].

Based on game theory, some scholars have studied the urban joint distribution to clarify the interest distribution and coordination among subjects. Hu et al. [20] discussed the influence of information sharing and benefit distribution on alliance based on evolutionary game theory. Peng et al. [21] proposed a two-stage income distribution strategy based on Nash bargaining method to solve the income distribution problem of task-oriented joint distribution alliance. Cheng et al. [22] used the Stackelberg game to solve the optimal revenue and service level of the express company, the terminal integration enterprise and the whole service supply chain in decentralized decision-making, and sought an allocation scheme that satisfied both the express company and the terminal integration enterprise. Wang et al. [23] believed that introducing logistics service providers as coordinators can facilitate the formation of cooperative alliances among distribution centers, which could effectively deal with the uneven distribution of cooperative benefits among distribution centers in the process of multi-center vehicle route optimization. Tao et al. [24] put forward the 2B/2C business integration and sharing model of urban food cold chain, and constructed the pricing game model of third-party logistics enterprises and fresh food e-commerce.

In summary, academic studies are mainly concentrated on the urban joint distribution study of route optimization and distribution mode [8–16,25–27], and there are also scholars researching the interests distribution between main participants, incentives and distribution mechanisms, from different perspectives [28–31], but there is little research on the role of the major player (namely, the government) in the city of joint distribution of in-depth discussion. It is of great significance to study the behavior mechanism of players and its evolution for the urban joint distribution alliance taking governments as the law regulators. Therefore, this paper constructs a cooperative model of multi-agent participation of urban joint distribution alliance from the perspective of government regulation. In the process of urban joint distribution, the government should take the leading role, and it is not only the initiator and direct participant of urban joint distribution projects, but also the coordinator of relevant interest agents. Meanwhile, it is the natural representative of the public and the supervisor of urban joint distribution. Under the condition of government rewards and punishments, what is the mechanism that urban joint distribution enterprises can reach an alliance? How will the behavior strategy of an urban joint distribution alliance evolve? Are there certain boundary conditions for government supervision of policy implementation? How to design regulations more quickly and efficiently to promote the strategy of urban joint distribution alliance? These problems deserve attention and deep thinking. Because of this, this paper established an urban joint distribution alliance cooperation evolutionary game model from the perspective of government regulation and took the influences of "stick and carrot" policy from the government on joint distribution alliance behavior strategies into consideration. On this basis, the urban joint distribution evolution rule and collaboration strategy alliance behaviors were discussed thoroughly, which has important theoretical and practical significance.

## 2. Theoretical Basis

In the famous "prisoner's dilemma" model of game theory, each prisoner eventually chooses a "betrayal" strategy to achieve a "Nash equilibrium", or non-cooperative equilibrium. In the "prisoner's dilemma", each player is egoistic, that is, they all seek the maximum self-interest, but do not care about the interests of another participant and the best choice of the group. In other words, in a group, the rational choice made by individuals leads to the collective irrationality [32–38]. Adam Smith's egoism believes that the process of pursuing individual interests is conducive to the effective promotion of

social interests, but the non-cooperative game theory challenges this view. In non-cooperative games, the opposite is true. The basic assumption of neoclassical economics is "rational man", while the reality is that, in many cases, human behavior or choices may be irrational because information is incomplete [39,40]. Due to constraints (such as information, etc.), individual choices are seemingly rational, but actually irrational from the perspective of collective rationality. Examples of such dilemmas in life include price competition, environmental protection, interpersonal relationships, etc.

The game relationship between urban joint distribution alliance enterprises is similar to the situation in the "prisoner's dilemma". Based on the maximization of their own interests and the minimization of costs, urban joint distribution enterprises make the decision that whether to participate in the urban joint distribution alliance. Individual rational choice may lead to collective irrationality. Eventually, urban joint distribution enterprises will only consider their own interests, instead of others and the overall interests, resulting in urban traffic congestion, ecological environment deterioration. The problem of urban joint distribution is a typical non-cooperative game problem. As an independent economic individual, the choice of urban joint distribution alliance enterprises seems rational at the first sight, but it is not reasonable from the perspective of the whole society. The addition of external execution mechanism will effectively promote the system to achieve the optimal solution. Therefore, the government's participation, coordination and guidance are very effective in solving the problem of cooperation of urban joint distribution alliance. Evolutionary game theory is a method that combines game and dynamic evolution. It can study the stable structure of game system and the strategy selection process of the behavior subject in the process of evolution by introducing dynamic mechanism. The basic idea is that in a group of a certain size, game players are not super rational players, and it is impossible to find the optimal equilibrium point in every game, but repeated games can achieve equilibrium through trial and error. Thus, the best strategy for game players is to imitate and improve the best strategy for themselves and others in the past. Through long-term imitation and improvement, all game players will tend to a certain stable strategy [41,42].

In the face of growing pressure of ecological environment and traffic congestion, government supervision to the urban joint distribution alliance behavior is around the corner. Obviously, the government regulations for behavior selection will affect the behavior selection of joint distribution enterprises. Therefore, a dynamic game process of joint distribution alliance behavior is formed, as is shown in Figure 1.

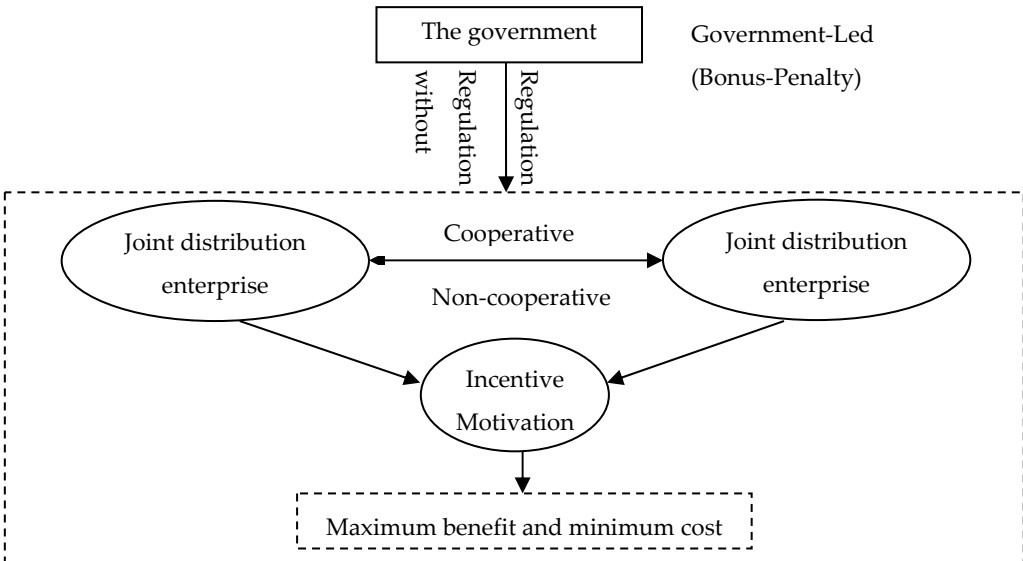

**Figure 1.** Dynamic game diagram of joint distribution alliance of government-supervised cities.

The multi-agent game relationship of urban joint distribution alliance is relatively complex, which includes both cooperation and competition. The factors influencing the strategy selection of

behavioral subjects involve many aspects. In this paper, literature search is conducted on the factors influencing the formation of urban joint distribution alliance, as is shown in Table 2.

**Table 2.** Table of influencing factors of urban joint distribution alliance.

| Author(s) (Year) [Reference] | Influencing factors |
|---|---|
| Zhang Rong (2011) [20] | Resource complementarity, resource utilization efficiency, enterprise reputation, alliance cooperation experience, interests distribution |
| Yin Mengmeng (2015) [43] | Cooperation excess return, input cost, risk cost, trust coefficient |
| Bai Shizhen (2015) [44] | Resource input, risk bearing cost, benefit distribution compensation, information sharing |
| He Mingke (2016) [45] | Complementary core resources and capabilities, degree of trust, information communication, operational risks, alliance profitability, profit distribution |
| Xian Chuanzhi (2016) [46] | Independent income, additional income from cooperation, income distribution coefficient, input cost, cooperation risk cost, government subsidies and penalties |
| Hu Jueliang et al. (2018) [47] | Independent income, information technology, enterprise cooperation cost, information sharing risk, cooperation additional income, income distribution coefficient |
| Hong Shuai et al. (2019) [48] | Cost allocation, benefit distribution, income distribution coefficient, cooperation additional income |
| Li Longxiao et al. (2019) [31] | Business operations, customer satisfaction, environmental sustainability, information technology |

According to the research achievements in recent years, it can be seen that the factors such as the independent income of urban joint distribution enterprises, extra income from cooperation, income distribution coefficient, enterprise cost (cost, risk cost, etc.), interests distribution compensation, the government's subsidies and penalty are affecting the formation of urban joint distribution enterprises. Therefore, according to the existing research results, this paper takes these major factors as the main parameters of the model to construct the evolutionary game model of the government-supervised city joint distribution alliance.

## 3. Model Construction

### 3.1. Research Hypothesis

Urban joint distribution is an integrated distribution service mode formed by the development of urban logistics to a certain stage. Multi-agents, including urban joint distribution enterprises and the government, have different interest demands for joint distribution. However, the fast and efficient operation of urban joint distribution depends on the behavioral strategy selection of each participant to maximize their own interests. Although the operation state of urban joint distribution enterprises is not quite the same, they can take advantage of their respective complementary advantages, and the enterprise alliance can promote each enterprise to maintain a certain collectivity. The effective and smooth implementation of urban joint distribution is a process in which the interests of enterprises interact with each other and participate in the game. In order to maximize the interests, enterprises are bound to choose behavioral strategies with their own interests in the game. Therefore, this paper first carries out model hypothesis and analysis on the evolutionary game among urban joint distribution enterprises.

**Hypothesis 1.** *Assume that the urban joint distribution alliance system is a complete ecosystem, and enterprises have bounded rationality and the ability to learn. They can change strategies through learning and ultimately maximize their own interests.*

**Hypothesis 2.** *Assume that the enterprises of the joint distribution alliance in 2 cities are homogeneous, and they all have two choices of strategic behavior. That is, they form alliance with other enterprises or conduct distribution for themselves. If $x, y$ $(x, y) \in [0, 1]$ are the probability that enterprise A and enterprise B choose cooperative strategy, and then $1 - x, 1 - y$ are the probability of non-cooperative strategy for enterprise A and enterprise B, respectively.*

Therefore, the evolutionary game payment matrix of enterprise A and enterprise B for urban joint distribution is constructed. The relevant parameters are assumed as follows: $\pi_1, \pi_2$ are the benefits that obtained by enterprise A and enterprise B respectively. $\Delta\pi$ is the synergetic benefit obtained by enterprise A and enterprise B when they implement the cooperative strategy. The proportion of synergetic benefit obtained by enterprise A is $\theta$, and the proportion of synergetic benefit obtained by enterprise B is $1 - \theta$. $c_1, c_2$ are the cost that enterprise A or enterprise B chooses cooperative behavior while the other party chooses individual behavior. $r_1, r_2$ are the external benefits obtained by the other enterprise due to the cooperative behavior of enterprise A or enterprise B. The income matrix between enterprise A and enterprise B is shown in Table 3.

**Table 3.** The profit matrix of collaborative cooperation between the two parties of urban joint distribution enterprises.

|  |  | Enterprise B | |
| --- | --- | --- | --- |
|  |  | Cooperation | Non-Cooperation |
| **Enterprise A** | **Cooperation** | $\pi_1 + \theta\Delta\pi - c_1, \pi_2 + (1 - \theta)\Delta\pi - c_2$ | $\pi_1 - c_1, \pi_2 + r_2,$ |
|  | **Non-Cooperation** | $\pi_1 + r_1, \pi_2 - c_2$ | $\pi_1, \pi_2,$ |

Due to a lack of system design, it is difficult for enterprises involved to reach agreement in the formation of urban joint distribution alliance for the sake of interest distribution. The government, as the leader of the urban joint distribution projects and the representative of public interests plays the cooperative and regulatory role in the implementation of the urban joint distribution.

The behavioral strategy selection of urban joint distribution enterprises are discussed from the perspective of government regulation. The government can choose to regulate or not regulate (the probability of supervision is $z$), and enterprises can still choose to cooperate or conduct distribution independently.

Firstly, the government is involved in regulation. If both enterprise A and enterprise B choose to conduct distribution on their own, they will certainly be punished by the government. Suppose the penalties for enterprise A and enterprise B are $k_1, k_2$, and the benefits of both parties herein are $\pi_1 - k_1, \pi_2 - k_2$ respectively. When enterprise A chooses cooperation strategy and enterprise B chooses non-cooperation strategy, enterprise A will pay the cost $c_1$ due to collaborative cooperation, but A will get incentive support $\Delta r_1$ from the government, and enterprise B will obtain external benefits $r_2$ due to cooperative strategy chosen by enterprise A. On the other hand, when enterprise B chooses collaborative strategy and enterprise A chooses to conduct distribution on its own, enterprise B will pay the cost $c_2$ for collaborative cooperation, but will get incentive support $\Delta r_2$ from the government. The regional economic benefit brought to the government is $\pi_3$ due to the implementation of urban joint distribution project, and the regulatory cost for enterprises is $c_3$.

Secondly, the government does not participate in regulation. Suppose that both enterprise A and enterprise B choose to conduct distribution on their own. Since the government does not regulate them, their profits will not be affected. When enterprise A chooses collaborative strategy and enterprise B chooses to conduct distribution on its own, enterprise A will pay the cost $c_1$ due

to collaborative cooperation, and enterprise B will obtain external benefits $r_2$ due to cooperative cooperation. When enterprise B chooses collaborative strategy and enterprise A chooses to conduct distribution on its own, enterprise B will pay the cost $c_2$ for collaborative cooperation, and enterprise A will obtain external benefits $r_1$ because enterprise B chooses collaborative cooperation. There are no benefits and regulatory costs for the government.

Therefore, this paper constructs a mixed strategy game matrix among urban co-distribution enterprises from the perspective of government regulation, as shown in Table 4.

**Table 4.** A mixed strategy game matrix among urban co-distribution enterprises from the perspective of government regulation.

| Strategy Set | Enterprise A | Enterprise B | Government C |
|:---:|:---:|:---:|:---:|
| (1,1,1) | $\pi_1 + \theta\Delta\pi - c_1$ | $\pi_2 + (1-\theta)\Delta\pi - c_2$ | $\pi_3 - c_3$ |
| (1,1,0) | $\pi_1 + \theta\Delta\pi - c_1$ | $\pi_2 + (1-\theta)\Delta\pi - c_2$ | 0 |
| (1,0,1) | $\pi_1 - c_1 + \Delta r_1$ | $\pi_2 + r_2 - k_2$ | $\pi_3 - c_3 - \Delta r_1 + k_2$ |
| (1,0,0) | $\pi_1 - c_1$ | $\pi_2 + r_2$ | 0 |
| (0,1,1) | $\pi_1 + r_1 - k_1$ | $\pi_2 - c_2 + \Delta r_2$ | $\pi_3 - c_3 - \Delta r_2 + k_1$ |
| (0,1,0) | $\pi_1 + r_1$ | $\pi_2 - c_2$ | 0 |
| (0,0,1) | $\pi_1 - k_1$ | $\pi_2 - k_2$ | $\pi_3 - c_3 + k_1 + k_2$ |
| (0,0,0) | $\pi_1$ | $\pi_2$ | 0 |

### 3.2. Equilibrium Analysis of Three-Party Evolutionary Game

Based on the above assumptions and payment matrix, the government and enterprises can obtain corresponding benefits by adopting different strategies and establish a replication dynamic system.

Suppose that the expected profit of enterprise A choosing "collaborative cooperation" is the mixed strategy game matrix among urban co-distribution enterprises from the perspective of government regulation in Table 4, then there is:

$$\begin{aligned} E_{A1} &= yz(\pi_1 + \theta\Delta\pi - c_1) + y(1-z)(\pi_1 + \theta\Delta\pi - c_1) + \\ &\quad (1-y)z(\pi_1 - c_1 + \Delta r_1) + (1-y)(1-z)(\pi_1 - c_1) \\ &= \pi_1 - c_1 + y\theta\Delta\pi + z\Delta r_1 - yz\Delta r_1 \end{aligned} \tag{1}$$

If enterprise A chooses non-cooperative strategy, its expected benefit is $E_{A2}$, then there is:

$$\begin{aligned} E_{A2} &= yz(\pi_1 + r_1 - k_1) + y(1-z)(\pi_1 + r_1) + (1-y)z(\pi_1 - k_1) + (1-y)(1-z)\pi_1 \\ &= \pi_1 + yr_1 - zk_1 \end{aligned} \tag{2}$$

$E_{A1}$ and $E_{A2}$ are actually the benefits obtained by enterprise A when it adopts different strategies. If the average expected benefit for enterprise A $\overline{E}_A$ is $\overline{E}_A = xE_{A1} + (1-x)E_{A2}$, then the replication dynamic equation of the probability of enterprise A choosing cooperation strategy is as follows:

$$\begin{aligned} F(x) &= \tfrac{dx}{dt} = x(E_{A1} - \overline{E}_A) \\ &= x(1-x)[-c_1 + y(\theta\Delta\pi - r_1) + z(\Delta r_1 + k_1) - yz\Delta r_1] \end{aligned} \tag{3}$$

The replication dynamic equation of probability of cooperative behavior strategy selected by enterprise B is as follows:

$$\begin{aligned} F(y) &= \tfrac{dy}{dt} = y(E_{B1} - \overline{E}_B) \\ &= y(1-y)\{-c_2 + x[(1-\theta)\Delta\pi - r_2] + z(\Delta r_2 + k_2) - xz\Delta r_2\} \end{aligned} \tag{4}$$

where $E_{B1}$ is the expected benefit for cooperation strategy of enterprise B, and $\overline{E}_B$ is the average expectation of benefits for enterprise B.

The replication dynamic equation of probability of regulation behavior strategy selected by the government is as follows:

$$
\begin{aligned}
F(z) &= \frac{dz}{dt} = z(E_{C1} - \overline{E}_C) \\
&= z(1-z)[\pi_3 - c_3 + k_1 + k_2 - x(\Delta r_1 + k_1) - y(\Delta r_2 + k_2) + xy(\Delta r_1 + \Delta r_2)]
\end{aligned}
\tag{5}
$$

where, $E_{C1}$ is the expected benefits for the regulative behavior strategy and $\overline{E}_B$ is the average expected benefits. The above analysis shows that if the benefits and payment cost generated when the government and enterprises choose non-cooperation strategy is less than the tripartite cooperation benefits and pay cost, three parties will choose cooperation strategy. There is $x = 1, y = 1, z = 1$, which means enterprise A and enterprise B choose cooperation strategy and government C choose positive regulation strategy, vice versa.

## 4. Results and Discussion

The stability of the evolutionary equilibrium point of urban joint distribution system consisting enterprise A, enterprise B, and the government are analyzed according to the different value range of parameters.

### 4.1. Evolutionary Stabilization Strategy of Enterprise A

The partial derivative of the replication dynamic equation of probability of enterprise A's choice of cooperation strategy $F(x)$ is obtained, then:

$$
\frac{dF(x)}{dt} = (1 - 2x)[-c_1 + y(\theta\Delta\pi - r_1) + z(\Delta r_1 + k_1) - yz\Delta r_1]
\tag{6}
$$

If $z = \frac{c_1 - y(\theta\Delta\pi - r_1)}{\Delta r_1(1-y)+k_1}$, then enterprise A in the 3d dynamic system is in A stable state; if $z > \frac{c_1 - y(\theta\Delta\pi - r_1)}{\Delta r_1(1-y)+k_1}$, then $x^* = 1$ is the evolutionary stabilization strategy; if $z < \frac{c_1 - y(\theta\Delta\pi - r_1)}{\Delta r_1(1-y)+k_1}$, then $x^* = 0$ is the evolutionary stabilization strategy.

In the three-dimensional dynamic system, the evolutionary stabilization strategy for enterprise A choosing cooperation strategy is shown in Figure 2 below.

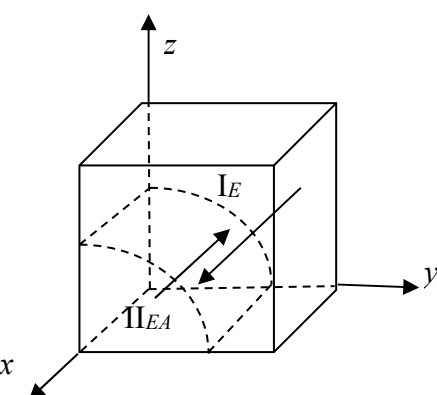

**Figure 2.** The evolutionary stabilization strategy for enterprise A choosing cooperative behavior strategy.

If $z = \frac{c_1 - y(\theta\Delta\pi - r_1)}{\Delta r_1(1-y)+k_1}$, the evolution strategy map is divided into two parts. One part is the volume $V_{I_{EA}}$ of the area IEA, which represents the probability that enterprise A chooses the cooperation strategy. The volume $V_{II_{EA}}$ of area IIEA represents the probability that enterprise A chooses the non-cooperation strategy.

**Proposition 1.** *The probability of enterprise A choosing cooperation strategy x is an increasing function to y, z, which meanwhile, is also an increasing function of government incentive income, cooperative income of urban joint distribution enterprises and government penalty, but it is a decreasing function of cost that enterprise A needs to pay when choosing "cooperative" behavior.*

**Proof of Proposition 1.** $\frac{dF(x)}{dt} = (1 - 2x)[-c_1 + y(\theta\Delta\pi - r_1) + z(\Delta r_1 + k_1) - yz\Delta r_1]$ is the partial derivative of the replication dynamic equation $F(x)$ of the cooperation strategy selected by enterprise A, and then the function of the probability of enterprise A choosing cooperation strategy $x$ and the probability of enterprise B choosing cooperation strategy $y$ is as follows:

$$x = \begin{cases} 0 & if & y < \frac{c_1 - z(\Delta r_1 + k_1)}{\theta\Delta\pi - r_1 - z\Delta r_1} \\ [0,1] & if & y = \frac{c_1 - z(\Delta r_1 + k_1)}{\theta\Delta\pi - r_1 - z\Delta r_1} \\ 1 & if & y > \frac{c_1 - z(\Delta r_1 + k_1)}{\theta\Delta\pi - r_1 - z\Delta r_1} \end{cases} \tag{7}$$

The function of the probability of enterprise A choosing cooperation strategy $x$ and the probability of government C choosing "regulative" behavior strategy $z$ is as follows:

$$x = \begin{cases} 0 & if & z < \frac{c_1 - y(\theta\Delta\pi - r_1)}{\Delta r_1(1-y) + k_1} \\ [0,1] & if & z = \frac{c_1 - y(\theta\Delta\pi - r_1)}{\Delta r_1(1-y) + k_1} \\ 1 & if & z > \frac{c_1 - y(\theta\Delta\pi - r_1)}{\Delta r_1(1-y) + k_1} \end{cases} \tag{8}$$

As is shown in Equation (7), if $y < \frac{c_1 - z(\Delta r_1 + k_1)}{\theta\Delta\pi - r_1 - z\Delta r_1}$, the ESS of enterprise A choosing cooperation strategy is $x = 0$, which means that when the probability of enterprise B choosing cooperative strategy is low to a certain degree, enterprise A also tends to choose non-cooperation strategy so as to combat enterprise B's "free-riding" behavior. However, when the probability of enterprise B choosing "cooperative" strategy is high enough, enterprise A will choose "cooperative" strategy in order to avoid government penalty and obtain cooperative benefits and government rewards.

As is shown in Equation (8), if $z < \frac{c_1 - y(\theta\Delta\pi - r_1)}{\Delta r_1(1-y) + k_1}$, that is, the probability of the active supervision of government C is low to a certain degree, enterprise A is often good at exploiting policy loopholes for its own interests, then enterprise A will choose non-cooperative strategy. When the probability of active supervision is at a high level, enterprises which choose non-cooperative strategy will be punished, and finally, they will turn to cooperative strategy, and then the probability of cooperative behavioral strategy evolution will be 1.

It can be shown from the relationship between $y$ and $\frac{c_1 - z(\Delta r_1 + k_1)}{\theta\Delta\pi - r_1 - z\Delta r_1}$, $z$ and $\frac{c_1 - y(\theta\Delta\pi - r_1)}{\Delta r_1(1-y) + k_1}$ that the probability x of cooperative strategy selection for enterprise A is the increasing function of incentive profits of governments $\Delta r_1$, the urban joint distribution enterprises cooperation yields $r_1$ and the government fines $k_1$. That is to say, enterprise A's non-cooperation strategy can result in the loss of incentive income from governments and benefits from joint distribution alliance. Sometimes, enterprise A may pay fines as well. Although there is no much loss on collaboration cost, the probability of enterprise A choosing non-cooperative strategy is still low because of opportunity cost it takes. In summary, enterprise A tends to choose cooperative strategy based on self-interest.

In conclusion, from the perspective of government regulation, in order to effectively stimulate the cooperation of urban joint distribution enterprises, the level of government regulation should be strengthened, the incentive bonus should be increased, and cooperative incomes for the urban joint distribution enterprises should be improved effectively. At the same time, the government should also increase the punishment for non-cooperative enterprises to play a certain role in supervision and deterrence.　□

*4.2. Evolution and Stabilization Strategy of Enterprise B*

Similar to enterprise A, the partial derivative of the replication dynamic equation $F(y)$ of probability of enterprise B's choice of cooperation strategy can be obtained as follows.

$$\frac{dF(y)}{dt} = (1-2y)\{-c_2 + x[(1-\theta)\Delta\pi - r_2] + z(\Delta r_2 + k_2) - xz\Delta r_2\} \tag{9}$$

If $z = \frac{c_2 - x[(1-\theta)\Delta\pi - r_2]}{\Delta r_2(1-x)+k_2}$, then enterprise B is in a stable state in the three-dimensional dynamic system. If $z > \frac{c_2 - x[(1-\theta)\Delta\pi - r_2]}{\Delta r_2(1-x)+k_2}$, then $y^* = 1$ is the evolutionary stabilization strategy. If $z < \frac{c_2 - x[(1-\theta)\Delta\pi - r_2]}{\Delta r_2(1-x)+k_2}$, then $y^* = 0$ is the evolutionary stabilization strategy.

In the three-dimensional dynamic system, the replication, dynamic evolution and stable strategy of probability of the behavior strategy of cooperative strategy selected by enterprise B are shown in Figure 3 below.

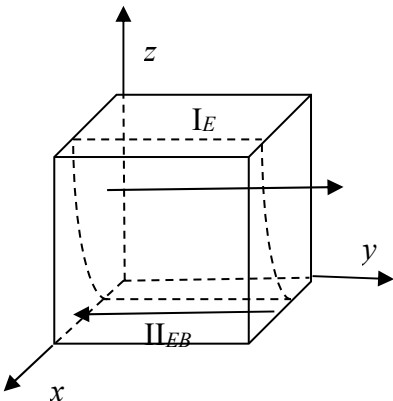

**Figure 3.** The dynamic evolution stable strategy graph of the probability of enterprise B choosing cooperative behavior strategy.

When $z = \frac{c_2 - x[(1-\theta)\Delta\pi - r_2]}{\Delta r_2(1-x)+k_2}$, the evolutionary strategy can be divided into two parts. $V_{I_{EB}}$, the area of IEB, represents the probability of enterprise B choosing cooperation strategy. $V_{II_{EB}}$, the area of IIEB, represents the probability of enterprise B choosing non-cooperation strategy.

**Proposition 2.** *The probability of enterprise B choosing cooperation strategy y is an increasing function of x, z. Meanwhile, it is also an increasing function of government incentive income, cooperative income of urban joint distribution enterprises and government penalty, but it is a decreasing function of cost that enterprise B needs to pay when choosing cooperation behavior.*

**Proof of Proposition 2.** Enterprise A and enterprise B are homogeneous enterprises, so enterprise B and enterprise A are in a symmetrical state, and their choice of cooperative behavior is similar to the analysis of enterprise A, which is not repeated here.   □

*4.3. Evolutionary Stabilization Strategy of Government C*

The partial derivative of the replication dynamic equation of probability $F(z)$ for government C choosing "positive regulation" behavior strategy can be obtained as follows:

$$\frac{dF(z)}{dt} = (1-2z)[\pi_3 - c_3 + k_1 + k_2 - x(\Delta r_1 + k_1) - y(\Delta r_2 + k_2) + xy(\Delta r_1 + \Delta r_2)] \tag{10}$$

If $x = \frac{\pi_3 - c_3 + k_1 + k_2 - y(\Delta r_2 + k_2)}{\Delta r_1 + k_1 - y(\Delta r_1 + \Delta r_2)}$, then government C in the 3d dynamic system is in a stable state. If $x > \frac{\pi_3 - c_3 + k_1 + k_2 - y(\Delta r_2 + k_2)}{\Delta r_1 + k_1 - y(\Delta r_1 + \Delta r_2)}$, then $z^* = 1$ is the evolutionary stabilization strategy. If $x < \frac{\pi_3 - c_3 + k_1 + k_2 - y(\Delta r_2 + k_2)}{\Delta r_1 + k_1 - y(\Delta r_1 + \Delta r_2)}$, then $z^* = 0$ is the evolutionary stabilization strategy.

In the three-dimensional dynamic system, the replication evolutionary stabilization strategy of government C choosing "active supervision" behavior strategy is shown in Figure 4 below:

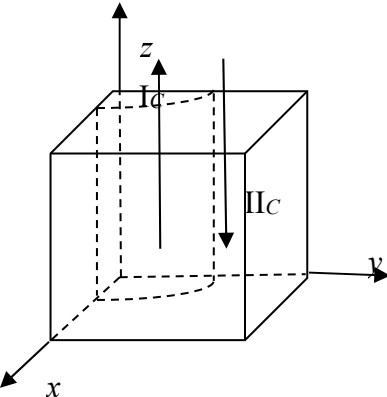

**Figure 4.** The dynamic evolution and stable strategy graph of probability of government C choosing active regulation behavior strategy.

When $x = \frac{\pi_3 - c_3 + k_1 + k_2 - y(\Delta r_2 + k_2)}{\Delta r_1 + k_1 - y(\Delta r_1 + \Delta r_2)}$, the evolutionary strategy diagram is divided into two parts. $V_{I_C}$, the area of IC, represents the probability of government C choosing positive regulation. $V_{II_C}$, the area of IIC, represents the probability of government C choosing positive regulation.

**Proposition 3.** *The probability z that government C chooses the behavioral strategy of "positive regulation" is a decreasing function of x, y. At the same time, the higher the regional economic benefits brought by the implementation of urban joint distribution projects, the lower the supervision cost for enterprises, and the more inclined the government is to choose "positive regulation". When enterprises choose non-cooperative strategy, the higher the punishment cost of the government grants. When enterprises choose cooperative strategy, the lower the incentive support of the government is, the more inclined the government is to choose "positive regulation" strategy in order to improve its own benefits.*

**Proof of Proposition 3.** The partial derivative of the replication dynamic equation of probability of government C's choice of "positive regulation" behavior strategy can be obtained as follows.

$$\frac{dF(z)}{dt} = (1 - 2z)[\pi_3 - c_3 + k_1 + k_2 - x(\Delta r_1 + k_1) - y(\Delta r_2 + k_2) + xy(\Delta r_1 + \Delta r_2)] \tag{11}$$

The function of probability of government C choosing "positive regulation" behavior strategy to the probability of enterprise A choosing cooperative behavior strategy is as follows:

$$z = \begin{cases} 0 & if \quad x > \frac{\pi_3 - c_3 + k_1 + k_2 - y(\Delta r_2 + k_2)}{\Delta r_1 + k_1 - y(\Delta r_1 + \Delta r_2)} \\ [0,1] & if \quad x = \frac{\pi_3 - c_3 + k_1 + k_2 - y(\Delta r_2 + k_2)}{\Delta r_1 + k_1 - y(\Delta r_1 + \Delta r_2)} \\ 1 & if \quad x < \frac{\pi_3 - c_3 + k_1 + k_2 - y(\Delta r_2 + k_2)}{\Delta r_1 + k_1 - y(\Delta r_1 + \Delta r_2)} \end{cases} \tag{12}$$

The function of probability of government C choosing "positive regulation" behavior strategy to the probability of enterprise B choosing cooperative behavior strategy is as follows:

$$
z = \begin{cases} 0 & if \quad y > \frac{\pi_3 - c_3 + k_1 + k_2 - x(\Delta r_1 + k_1)}{\Delta r_2 + k_2 - x(\Delta r_2 + \Delta r_1)} \\ [0, 1] & if \quad y = \frac{\pi_3 - c_3 + k_1 + k_2 - x(\Delta r_1 + k_1)}{\Delta r_2 + k_2 - x(\Delta r_2 + \Delta r_1)} \\ 1 & if \quad y < \frac{\pi_3 - c_3 + k_1 + k_2 - x(\Delta r_1 + k_1)}{\Delta r_2 + k_2 - x(\Delta r_2 + \Delta r_1)} \end{cases} \tag{13}
$$

It can be seen from the Equations (12) and (13) that when $x > \frac{\pi_3 - c_3 + k_1 + k_2 - y(\Delta r_2 + k_2)}{\Delta r_1 + k_1 - y(\Delta r_1 + \Delta r_2)}$ or $y > \frac{\pi_3 - c_3 + k_1 + k_2 - x(\Delta r_1 + k_1)}{\Delta r_2 + k_2 - x(\Delta r_2 + \Delta r_1)}$, the ESS for the "positive regulation" of government C is $z = 0$, which means if the probability of cooperative strategy chosen by enterprises is large, the government does not take great efforts to the regulation. Therefore, the ESS of government C choosing "positive regulation" strategy is $z = 0$. However, when $x < \frac{\pi_3 - c_3 + k_1 + k_2 - y(\Delta r_2 + k_2)}{\Delta r_1 + k_1 - y(\Delta r_1 + \Delta r_2)}$ or $y < \frac{\pi_3 - c_3 + k_1 + k_2 - x(\Delta r_1 + k_1)}{\Delta r_2 + k_2 - x(\Delta r_2 + \Delta r_1)}$, the ESS for the "positive regulation" of government C is $z = 1$, which means when enterprises are unwilling to cooperate in urban joint distribution projects, the supervision should be strengthened to realize the implementation of urban joint distribution projects.

It can be seen from the relationship between $x$ and $\frac{\pi_3 - c_3 + k_1 + k_2 - y(\Delta r_2 + k_2)}{\Delta r_1 + k_1 - y(\Delta r_1 + \Delta r_2)}$, $y$ and $\frac{\pi_3 - c_3 + k_1 + k_2 - x(\Delta r_1 + k_1)}{\Delta r_2 + k_2 - x(\Delta r_2 + \Delta r_1)}$ that the probability of "positive regulation" strategy $z$ is an increasing function of the urban regional economic benefits of the implementation of the joint distribution projects $\pi_3$ and the government fines $(k_1, k_2)$. At the same time, as the regulatory costs of the enterprise $c_3$ and rewarding support from government $(\Delta r_1, \Delta r_2)$ decrease, the government regulation will be strengthened, and the ESS of the government is stable in 1.

In conclusion, the effective implementation of urban joint distribution projects is a process of collaborative cooperation between enterprises under the regulation of the government. As long as both enterprises have the self-discipline of active cooperation, the probability of government supervision is relatively small. In the actual supervision process, the regional economic benefits brought by the implementation of the urban joint distribution project and the larger the government penalty when the enterprises do not cooperate will, drive the government to intensify its supervision in order to increase its economic benefits and establish the public image of the government. The smaller cost of active government supervision and the incentive support for enterprises will also help, that is to say, the smaller the government input, the more active it will be to adopt supervision. □

*4.4. Strategy Comparison of Urban Joint Distribution Enterprises from the Perspective of Working with Government Regulations or without Government Regulation*

According to the above analysis, when the government regulates the urban joint distribution project, it will promote the formation of the urban joint distribution alliance project. If the project is implemented, the government will not participate in the regulation. At this time, the replication dynamic equations of urban co-distribution enterprises A and B are as follows:

$$
\frac{d\overline{F}(x)}{dt} = (1 - 2x)[-c_1 + y(\theta \Delta \pi - r_1)] \tag{14}
$$

$$
\frac{d\overline{F}(y)}{dt} = (1 - 2y)\{-c_2 + x[(1 - \theta)\Delta \pi - r_2]\} \tag{15}
$$

From the comparison of the replication dynamic equation of urban co-distribution enterprises with or without government regulations, it can be seen that:

$$
\frac{dF(x)}{dt} - \frac{d\overline{F}(x)}{dt} = (1 - 2x)[z(\Delta r_1 + k_1) - yz\Delta r_1] \tag{16}
$$

$$\frac{dF(y)}{dt} - \frac{d\overline{F}(y)}{dt} = (1 - 2y)[z(\Delta r_2 + k_2) - xz\Delta r_2] \tag{17}$$

That is to say, when the government participates in the urban joint distribution alliance, it will improve the average profits for the enterprises by $z(\Delta r_1 + k_1) - yz\Delta r_1$ and $z(\Delta r_2 + k_2) - xz\Delta r_2$. According to relevant assumptions, $z(\Delta r_1 + k_1) - yz\Delta r_1 > 0$ and $z(\Delta r_2 + k_2) - xz\Delta r_2 > 0$, it means when the government participates in the project of urban joint distribution, it will increase the average income of enterprises and promote the formation of urban joint distribution alliance. Therefore, a reasonable government subsidy and incentive regulation mechanism is the key to the formation of urban joint distribution alliance.

## 5. Stimulation Analysis

In 2013, the Chinese Ministry of Commerce selected the city of Urumqi as a joint distribution pilot city, which was in the position of speeding up the construction and development of city logistics distribution system together. For instance, Xinjiang Baicheng Investment Co. Ltd. (Baicheng for short) and Urumqi Mingcheng Zhonghe Logistics Co. Ltd. (Mingcheng for short) received government encouragement and support for participation in urban joint distribution project.

The hypothesis for the simulation model parameters is set as follows: the basic earning of Baicheng is 20 million RMB Yuan/year, while the basic earning of Mingcheng is 25 million RMB Yuan/year. If both enterprises choose cooperative strategy, the collaborative earning will be 10 million RMB Yuan/year, and the earnings ratio is 5:5. In order to promote the urban joint distribution project, Baisheng invested 12 million RMB Yuan per year, and Mingsheng invested 15 million RMB Yuan per year. If only one enterprise chooses cooperative strategy, the other enterprise will also get an external benefit of 1 million RMB Yuan/year due to the improvement of logistics environment. The government grants 10 million RMB Yuan for supporting the urban joint distribution project. When both enterprises choose cooperative strategies, they will receive 4 million RMB Yuan and 6 million RMB Yuan from the government respectively. However, once market competition between two enterprises was found, the two enterprises will be fined 1.2 million RMB Yuan and 1.5 million RMB Yuan respectively. As a result of the cooperation between the two enterprises, the regional economic benefit growth and logistics environment improvement brought to the government is approximately 10 million RMB Yuan/year, and the cost for the government carrying out the platform construction and other costs for the project supervision is 35 million RMB Yuan/year.

(1) Probability of enterprises' cooperation strategy selection with or without government's participation in urban joint distribution projects

It can be seen from the probability of enterprises' cooperation strategy selection with or without government's participation in urban joint distribution projects that (Figure 5): when the government does not participate in joint distribution projects, namely, no government incentive nor supervisory mechanism, due to the rational man hypothesis, even if there is strong cooperation willingness at the beginning, then it will be driven by benefit maximization, and cause the failure of cooperation. When the government actively participates in the project of urban joint distribution, it will set certain incentive and regulation mechanism to promote the cooperation among enterprises and realize reasonable distribution of logistics resources.

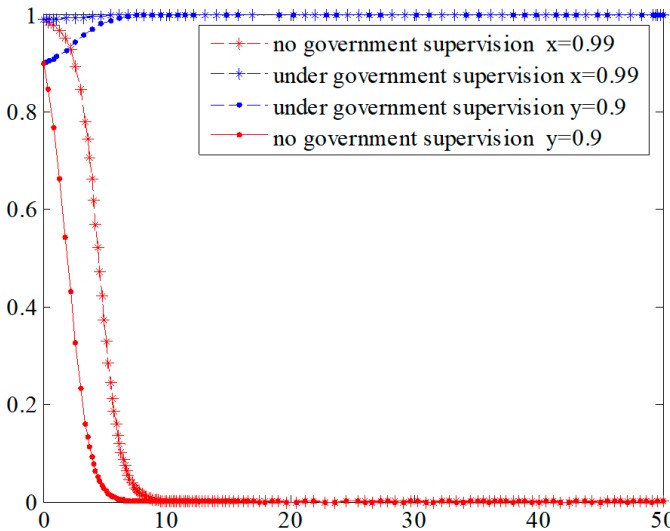

**Figure 5.** The probability of enterprises' cooperation strategy selection with or without government regulation.

(2) Analysis of the relationship between the probability of cooperation strategy and positive regulation

When the urban joint distribution project is carried out, different combination of strategies between enterprises and government will affect the final evolution result. When government regulation is strict, enterprises will suffer high opportunity cost once they choose non-cooperation strategy, and then they tend to choose cooperation strategy. However, when the government supervision is relatively loose, Baisheng and Mingsheng may choose opportunistic behaviors, and they tend to choose non-cooperative strategy. It is shown from Figure 6 that the stricter the government regulation is, the greater the larger the probability of cooperation, which in turn is to promote regional economic growth and improve earnings logistics environment and finally form a benign logistics environment.

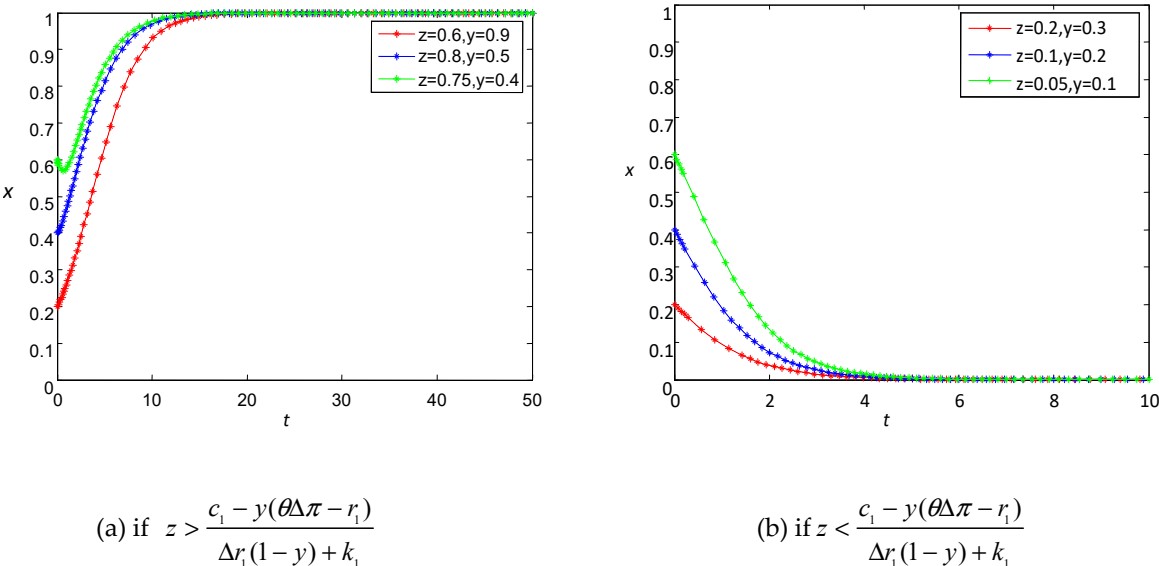

$$\text{(a) if } z > \frac{c_1 - y(\theta\Delta\pi - r_1')}{\Delta r_1'(1-y) + k_1}$$

$$\text{(b) if } z < \frac{c_1 - y(\theta\Delta\pi - r_1')}{\Delta r_1'(1-y) + k_1}$$

**Figure 6.** Relationship between the evolutionary stability probability of enterprise A, and y and z.

(3) The influences of punishment and supportive rewards reduction on tripartite cooperation

Promotion of the urban joint distribution project aims to form a virtuous cycle of logistics environment and the market mechanism. Therefore, the government will exit regulation at a certain

stage of project implementation. The influences of penalties and supportive rewards reduction on the urban joint distribution project are as shown in Figure 7.

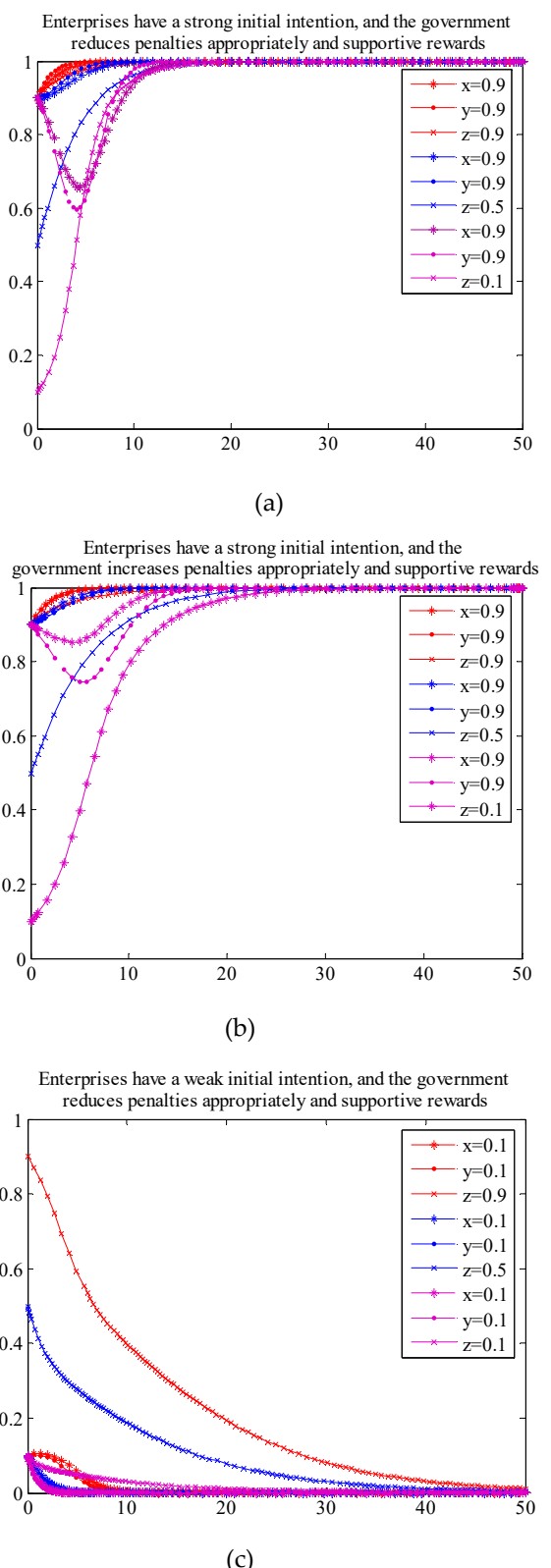

**Figure 7.** The contrast chart with the strong and weak initial intention.

The reduction of the penalties and supportive rewards has a negative correlation with the evolution of the enterprises and the local economy (interests of government). When enterprises can fulfill their corporate social responsibility in accordance with the requirements for the government and to improve the cooperative willingness, even if the government reduces penalties appropriately and supportive rewards, it will not affect the implementation of the urban joint distribution projects. Once the benign logistics environment and market mechanism is formed, government can gradually withdraw its regulation. However, if the initial cooperation intention of enterprises is not strong, once the government reduces the punishment and supportive rewards, the final cooperation of urban joint distribution project will fail. It indicates that government regulation is of great significance in the initial stage of urban co-distribution project, otherwise there may be non-cooperation strategies. However, once the project is mature and enterprises actively choose cooperative strategy with higher profits, even if the government gradually withdraws from supervision, it will not have a significant impact on the final cooperation.

(4) Cost reduction and earnings improvement on tripartite collaboration

At the beginning of the project implementation, enterprises will spend a lot of money to develop infrastructure. At this time, the cost is relatively high. When the investment cost is greater than its earnings and supporting reward, no matter how well the government's willingness is (the willingness from 0.1 to 0.9), the enterprises may eventually choose non-cooperation strategies. The government should grant more supportive incentives to encourage enterprises to choose cooperation strategies. When the implementation of urban joint distribution project reaches a certain stage, the cost of the enterprise will decrease, and the enterprise will take the initiative to choose cooperation strategy. The impact of enterprises' cost input on tripartite collaborative cooperation is shown in Figure 8. Increasing the cooperative earnings also has a significant influence on tripartite cooperation. Therefore, how to improve collaborative benefits of enterprises in the process of project implementation should be given priority. Once the collaborative earnings of enterprises are improved, government can gradually withdraw from project supervision, which has little influence on the collaborative cooperation, as is shown in Figure 9.

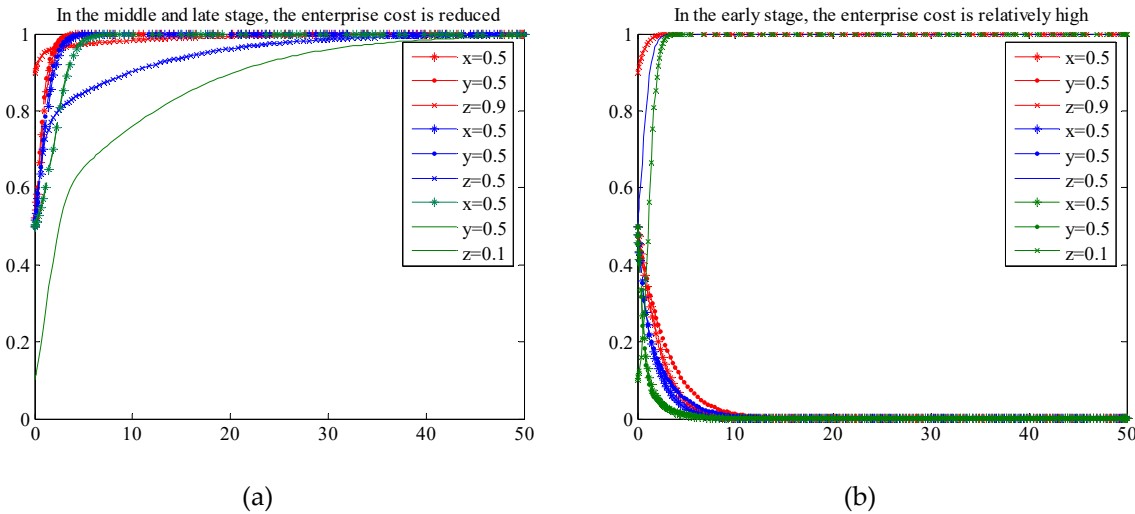

**Figure 8.** Impact of enterprise cost on evolution results.

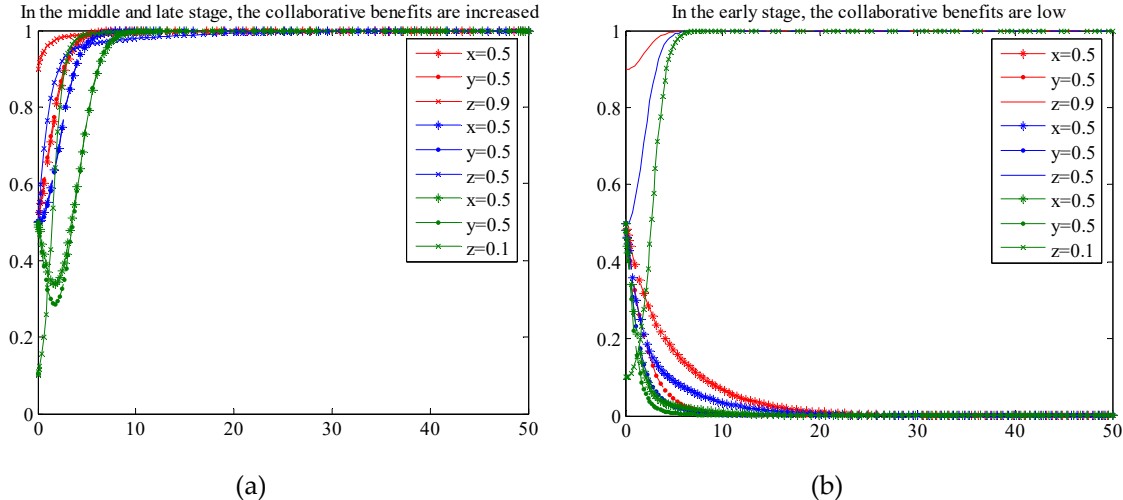

**Figure 9.** Influence of collaborative benefits on evolutionary results.

## 6. Research Conclusions and Management Implications

### 6.1. Research Conclusions

In order to effectively solve the practical problems in the process of the urban joint distribution management, the paper, from the perspective of the crucial role of government regulations, discusses the urban joint distribution alliance regulation mechanism and associated evolution of behavior. With enterprises as the main body and the introduction of government regulation parameters, the urban joint distribution model was built based on evolutionary game method. The alliance behavior mechanism and evolution laws are analyzed based on the relationship between the parameters and the simulation analysis, and then the alliance behavior strategy selection is analyzed from the perspective of government supervision. At the same time, the actual data is applied in the city joint distribution alliance cooperation game model for empirical validation. The solution of the problem plays an important guiding role in the process of project implementation and the proposed countermeasures and suggestions wherein are of great significance.

It was found in the study that firstly, from the perspective of government regulation, at the beginning of the implementation of urban joint distribution project, when the profit of cooperation between enterprise alliances is low and the cost is relatively high, it is urgent for the government to formulate incentive policies to improve the government subsidies or increase the punishment for non-cooperation. Once a benign logistics environment and market mechanism is formed, the benefits of cooperation between enterprise alliances are increased and costs are reduced, and the government can withdraw from supervision in due course. Secondly, the urban joint distribution enterprises operate under the government regulations in the process of the game, the enterprises would actively cooperate with the alliance and receive government subsidies rather than passively accept government regulation. In order to obtain the sustainable development of the logistics of the ecological environment and the market mechanism, the enterprise should not only consider their own basic income but also accept government regulation.

### 6.2. Management Implications

From overall perspective: it is necessary to establish a government regulation for the sustainable development of the urban joint distribution alliance and build an effective incentive and supervision mechanism, so as to conduct long-term planning and overall optimization. Evaluation on the costs and benefits is the key to ensure the cooperation relationship between the government and enterprises. When the benefits of the government and enterprises exceed the costs, the game players will be encouraged to cooperate and form a joint distribution alliance, so as to maximize the cooperative benefits

of the joint distribution alliance and realize a win-win situation between the government and enterprises. From the government's perspective: on the one hand, there are many management implications from the perspective of governments. First of all, the government can give more policy support or financial support to enterprises in the early stage of promoting urban joint distribution projects, which can promote enterprise alliances to cooperate and form a benign logistics environment, and thus reduce the government's regulatory costs. Secondly, the government can make enterprises passively accept cooperation by doubling the penalty, which will increase the cost of non-cooperation, and reduce their non-cooperation willingness. Once the cooperation mechanism is formed, the government can adjust the policy appropriately to avoid affecting the cooperative behavior tendency of enterprise alliance. Finally, in the middle and late stage of project implementation, when the enterprise alliance has been formed, the government's regulatory role is not obvious. At this time, the government can choose to gradually withdraw in order to reduce the regulatory cost.

There are many management implications from the perspective of enterprises. First, enterprises can optimize the logistics network, improve the level of logistics technology and information technology, enhance the coordination of joint distribution of enterprise alliances, and realize the efficient and seamless link of enterprise alliances in information resource sharing, so as to reduce the cost of joint distribution of cities and improve the efficiency of joint distribution of cities. Second, the enterprise should guarantee to gradually improve the cooperative earnings of the alliance without increasing the cost, formulate the agreement and coordination mechanism conducive to the cooperative cooperation of the enterprise alliance, promote the long-term cooperative cooperation of the enterprise alliance, improve the economic income of the enterprise, and gradually develop towards a more professional direction.

**Author Contributions:** Conceptualization, N.Z. and Y.Y.; methodology, N.Z.; software, X.Z.; validation, N.Z., X.Z. and Y.Y.; investigation, X.Z.; resources, N.Z.; data curation, N.Z.; writing—original draft preparation, N.Z.; writing—review and editing, N.Z.; visualization, N.Z.; supervision, Y.Y.; project administration, N.Z.; funding acquisition, N.Z.

**Funding:** The relevant researches done in this paper are supported by the Natural Science Foundation of China (Grant No. 41801119), the Royal Society and NSFC International Exchanges project (IEC\NSFC\170391), Social Science Foundation of China (Grant No. 18FGL003), Key Project of National Language Commission (ZDI135-67), China Postdoctoral Science Foundation funded project (Grant No. 2018M631220), Excellent Youth Foundation of Xinjiang Scientific Committee (Grant No. 2017Q071), Foundation of Shihezi University (RCSX201754).

**Conflicts of Interest:** The authors declare no conflict of interest. The funders had no role in the design of the study; in the collection, analyses or interpretation of data; in the writing of the manuscript or in the decision to publish the results.

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
