# Peer review of "The Behavior Mechanism of the Urban Joint Distribution Alliance under Government Supervision from the Perspective of Sustainable Development"

_sustainability, doi:10.3390/su11226232_

Round 1

Reviewer 1 Report

As far I can understand the basic argument is sound in that government can increase incentives to cooperate among the firms by improving urban city logistics and reverting non-cooperative games into cooperative ones. There should be more discussion of the overall performance consequences of such actions. How can we be certain that government induces subsidizes are really worthwhile in the overall development of the cities.

Apart for the many actual figures in the manuscript the figure 1 is not an actual figure. Therefore, I advice to make it as a table or a list of things.

For those not experts in game theory this is a hard read with the multitude of equations. Further explanation of the formulas would be much appreciated to make the results more readable to larger audiences.

Moreover, there is still room for more elaboration in the conclusions.  A practical advice in this respect is the evaluation of the cost and benefits for all the parties involved, Firms, government, local government and citizens.

Author Response

Authors’ response

#1 reviewer

Points1: As far I can understand the basic argument is sound in that government can increase incentives to cooperate among the firms by improving urban city logistics and reverting non-cooperative games into cooperative ones. There should be more discussion of the overall performance consequences of such actions. How can we be certain that government induces subsidizes are really worthwhile in the overall development of the cities.

Response 1: The influences of government involvement in the urban joint distribution on the cooperation between enterprises and on the formation of joint distribution alliance are discussed, as shown in the result and discussion part. The contrast of probability for enterprises collaborating with or without government’s involvement is conducted as well. It indicates that government subsidies and incentive regulation mechanism are necessity in the formation of an urban joint distribution project(please refer to the numerical simulation and discussion part).

Points 2: Apart for the many actual figures in the manuscript the Figure 1 is not an actual figure. Therefore, I advice to make it as a table or a list of things.

Response 2: Figure 1was changed into a table, and the relevant regulations on urban joint distribution are also added. Thanks for the valuable suggestion.

Points 3: For those not experts in game theory this is a hard read with the multitude of equations. Further explanation of the formulas would be much appreciated to make the results more readable to larger audiences.

Response 3: The theoretical background of evolutionary game is introduced in part 2, and some explanations of the evolutionary game formula in the equilibrium analysis are provided as well.

Points 4: Moreover, there is still room for more elaboration in the conclusions.  A practical advice in this respect is the evaluation of the cost and benefits for all the parties involved, Firms, government, local government and citizens.

Response 4: When the benefits exceed the cost, the game players will choose cooperative strategy and form the urban joint distribution alliance. Therefore, it is very important to evaluate the costs and benefits of multiple players. The relevant practical recommendations have been added in the Part 7.

Reviewer 2 Report

The article presents a relatively large section introduction. Here is a moderately inappropriate WordArt ”picture where are mentioned a list of regulations and documents on urban joint distribution since 2011. This "Introduction" section lists citations of 31 literary sources, but only 30 of them are listed in References. In the following section,“ Problem description and model Assumption", authors go straight to the research hypotheses. The article lacks the Theoretical Background section. The only references to literature are given only in the Introduction section. The article gives the impression that the authors create all the theories used in the text.

Practical application is an interesting benefit of the article, but unfortunately, the article is not written sufficiently well. I recommend introducing the article into a standard structure, including sections Theoretical Background, Methods, Results, and Discussion. Also, authors should make minor linguistic text edits and significant formatting edits.
For example:
- The form of diagrams of Figures 7 to 10 is not the happiest as the text overlaps.
- The equation on line 251 does not have all parentheses enclosed.
- There are letters in the bibliography to indicate the type of publication; I believe that this is not a standard citation style.

Author Response

Authors’ response

#2 reviewer

Points 1: The article presents a relatively large section introduction. Here is a moderately inappropriate WordArt ”picture where are mentioned a list of regulations and documents on urban joint distribution since 2011. This "Introduction" section lists citations of 31 literary sources, but only 30 of them are listed in References. In the following section,“ Problem description and model Assumption", authors go straight to the research hypotheses. The article lacks the Theoretical Background section. The only references to literature are given only in the Introduction section. The article gives the impression that the authors create all the theories used in the text.

Response 1: In accordance with the reviewer’s suggestion, Fig.1 was changed into the form of table, and the normative documents on urban joint distribution were added to highlight the importance of urban joint distribution in the process of sustainable development of cities, as is shown in table 1. References were added, as is seen in reference list 31. The theoretical background and basis for strategy selection were enriched in the manuscript, as is shown in Part 2.

Points2: Practical application is an interesting benefit of the article, but unfortunately, the article is not written sufficiently well. I recommend introducing the article into a standard structure, including sections Theoretical Background, Methods, Results, and Discussion. Also, authors should make minor linguistic text edits and significant formatting edits.

For example:

- The form of diagrams of Figures 7 to 10 is not the happiest as the text overlaps.

- The equation on line 251 does not have all parentheses enclosed.

- There are letters in the bibliography to indicate the type of publication; I believe that this is not a standard citation style.

Response 2: The paper structure was adjusted according to the modification requirements, and some format editing was conducted. For instance, figure7, figure8, figure 9, and figure 10 were redrawn. The relevant equations were double checked and line 251 was modified. The reference format was standardized. In addition, this paper has been improved and modified by Mr. Yang Yingjie, a tenured professor of Demontfort University.

Round 2

Reviewer 2 Report

I thank the authors for editing the submitted text. The article has been improved.